# Preparation and Combustion Mechanism of Boron-Based High-Energy Fuels

Likun Han [1,†], Runde Wang [2,†], Weiyi Chen [1], Zhe Wang [1,*], Xinyu Zhu [2] and Taizhong Huang [2,*]

1   College of Weaponry Engineering, Naval University of Engineering, Wuhan 430000, China
2   School of Chemistry and Chemical Engineering, University of Jinan, Jinan 250022, China
*   Correspondence: hlkwzz@163.com (Z.W.); chm_huangtz@ujn.edu.cn (T.H.); Tel.: +86-531-89735868 (T.H.)
†   The authors contributed equally to this work.

**Abstract:** Due to the characteristics of high energy density and a high calorific value, boron has become a high-energy fuel and shows great potential to be a high-performance candidate for propellants. However, the wide applications of boron are still limited by the characteristics of easy oxidization, ignition difficulty, a long combustion duration, and combustion products that readily adhere to the surface and inhibit full combustion. Therefore, how to overcome the shortcomings and improve the combustion efficiencies of boron-based fuels have become the highlights in exploring novel high-performance energetic materials. In this paper, the prevalent preparation methods and the corresponding combustion mechanisms of boron-based energetic materials are briefly summarized. The results showed that the boron-based energetic materials can be prepared by surface coating, mechanical milling, and ultrasonic mixing methods. At the same time, the corresponding ignition delay and combustion efficiency were also analyzed according to different combustion tests. The results showed that the boron-based composites with different additives had different combustion characteristics. The combustion of boron-based energetic materials can be optimized by removing surface oxide layers, providing extra heat, inhibiting the formation of or the rapid removal of the combustion intermediates, and increasing the diffusion rate of oxygen. With the improvement of the combustion efficiency of boron-based energetic materials, boron-based high-energy fuels will become more and more widely adopted in the future.

**Keywords:** boron; combustion; combustion efficiency; boron-based composite





## 1. Introduction

Aerospace aircraft and tactical missiles generally use ramjets or scramjets as the propulsion system to achieve long flight distances and high speeds [1]. At present, the potential performance of ramjets or scramjets in high-speed vehicles has been fully studied and reported [2]. Among all of the reported fuels, boron has a mass calorific value of 58 MJ/kg and a volume calorific value of 131 MJ/L. Some studies showed that boron-based solid propellant had a high burning rate and high atmospheric pressure at high temperature [3], which make boron-based fuels one of the most attractive materials for ramjet fuels and other propellants [2,4–9]. Through the mixing of boron with other reagents to prepare compounds, composites or even blended mixtures with improved combustion efficiency of boron-based energetic materials have attracted sustained attention [10]. In addition to the preparation of boron-based composites, the combustion procedure and combustion mechanism of boron-based materials, especially the particles, have been simulated. Kalpakli et al. and Chen et al. proposed models for the combustion procedure [11,12]. They separately proposed the competitive reaction model of boron–oxygen–polymer ((BO)n) consumption in the ignition stage, as well as boron evaporation and the boiling process in the combustion stage. A typical combustion model is presented in Figure 1.

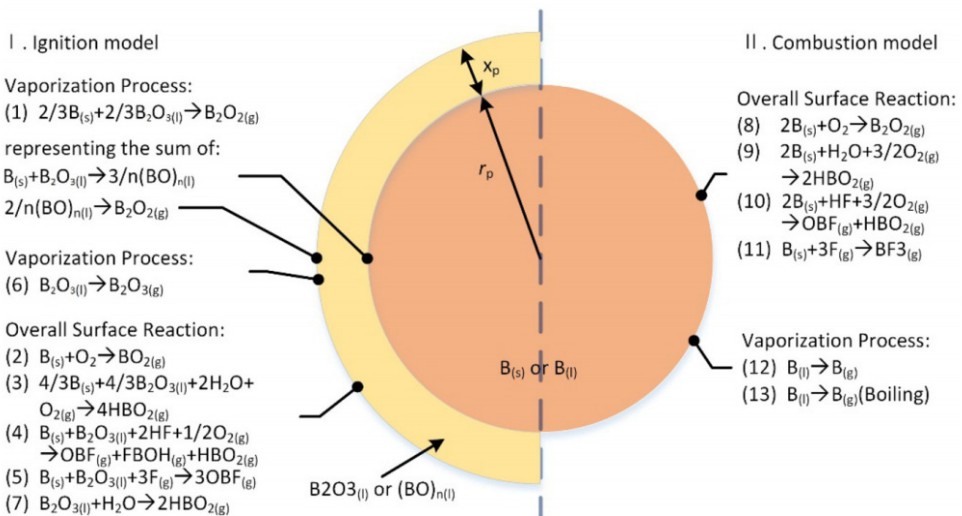

**Figure 1.** The ignition and combustion model of boron particles [11]. Reprinted with permission from Ref. [11] Fuel Processing Technology 2017.

As shown in Figure 1, the model fully took into account the evaporation process of $B_2O_3$, which can occur at 1860 °C, and $B_2O_2$. The reaction between boron oxide and water vapor in the ignition stage and the limitation of the oxygen molecular fraction, i.e., reactions 3 and 9, are the combustion rate-determining steps.

The equilibrium equation of boron particles in the ignition stage includes the following Equations (1)–(3) [13]:

$$\frac{dr_B}{dt} = -(\frac{2}{3}R_1 + R_2 + \frac{4}{3}R_3 + R_4 + R_5)\frac{M_B}{\rho_B} \tag{1}$$

$$\frac{dx_B}{dt} = -(\frac{2}{3}R_1 + \frac{4}{3}R_3 + R_4 + R_5 + R_6 + R_7)\frac{M_{B_2O_3}}{\rho_{B_2O_3}} \tag{2}$$

$$\frac{dE_B}{dt} = \frac{\partial E}{\partial T_B}\frac{dT_B}{dt} + \frac{\partial E}{\partial r_B}\frac{dr_B}{dt} + \frac{\partial E}{\partial x_B}\frac{dx_B}{dt}$$
$$= 4\pi(r_B + x_B)^2[-\sum_{i=1}^{5} R_i\Delta H_{R_i}^{298} + h_B(T_\infty - T_p) + \sigma_{B_2O_3}\varepsilon_{B_2O_3}(T_\infty^4 - T_B^4)] \tag{3}$$

where $E_B = (4/3\pi r_p^3 \rho_B c_{p,B} + 4\pi r_p^2 x_p \rho_{B_2O_3} c_{p,B_2O_3})$, $T_B$ is the energy of boron particles, $r_B$ is the radius of boron particles, $x_B$ is the thickness of the oxide layer, $R$ is the reaction rate, $M$ is the molecular weight, $h$ is the convective heat transfer coefficient, $\sigma$ is the Stefan-Boltzmann constant, $\varepsilon$ is the emissivity, and $c_p$ is the specific heat. The subscript of the number indicates the reaction as shown in Figure 1. Parameters can be found in reference [11].

The equilibrium equation of boron particles in the combustion stage is also calculated according to the following equations [13]:

$$\frac{dr_B}{dt} = -(\sum_{i=8}^{13} R_i)\frac{M_B}{\rho_B} \tag{4}$$

$$\frac{dE_B}{dt} = 4\pi r_B^2[\sum_{j=8}^{13} R_j\Delta H_{B,R_j}^{298} + h_B(T_\infty - T_B) + \sigma_B\varepsilon_B(T_\infty^4 - T_B^4)] \tag{5}$$

Here, $E_B = (4/3\pi r_B^3 \rho_B c_{p,B}) \cdot T_B$, $\Delta H_{B,R_i}^{298}$ is the standard enthalpy of reaction (i) per molecule of molecular boron.

The combustion processes of boron particles are divided into two stages, namely, the ignition stage and combustion stage [14–17]. The first stage involves the removal

of the oxide layer on the boron particle surface [18], and the second stage involves the combustion of boron particles [19]. $B_2O_3$ has a low melting point (450 °C) and a high boiling point (1860 °C). Liquid $B_2O_3$ has high viscosity and low mobility, making it easy to wrap the boron particles during the combustion process and eventually hinder the diffusion of surrounding oxygen, decreasing the combustion efficiency [20,21]. Therefore, the combustion of boron is easily reduced by the adhesive oxide layer on the surface. On the other hand, the combustion product of $B_2O_3$ usually has a high viscosity state and readily adheres to the boron surface accompanied by combustion, which eventually inhibits the complete combustion of boron particles. At the same time, the oxide layer that forms on boron particles also increases the difficulty regarding further ignition and burning [22].

During the burning process, $B_2O_3$ acts as an oxygen diffusion barrier and prevents the further combustion of boron particles [23,24]. When the combustion atmosphere temperature is high enough, $B_2O_3$ starts to evaporate and the boron particles can receive enough $O_2$ and keep burning. To overcome the barrier effect of $B_2O_3$ on combustion, different additives are introduced to increase the removal of $B_2O_3$ and improve the combustion efficiency of the boron particles. Different additives play different roles during the different stages of boron combustion. It has been proved that additives, such as metals [14,25–29], metal oxides [30–34], metal fluorides [35–38], fluoropolymers [23,39], nanocarbons [40–42], and energetic materials [21], can enhance the boron combustion efficiency. Combustion performance tests, such as the ignition delay time, combustion efficiency, ignition temperature, and combustion temperature, have also conducted, and the results were adopted to evaluate the combustion performance of boron. The structures of boron-based composites are usually characterized by scanning electron microscopy (SEM) and X-ray diffraction (XRD) tests.

In this paper, we reviewed the research on the preparation methodologies and combustion mechanism of boron-based energetic materials. Based on this review, we expect to present some beneficial advice on how to promote the preparation of boron-based energetic materials and deepen the corresponding understanding of combustion mechanisms.

## 2. Preparation of Boron-Based Fuels

Until now, the reported preparation methods of boron-based composites mainly include surface coating [21,43], mechanical milling [14,27,44], ultrasonic mixing [31], ultrasonic dispersion [41], high-energy mechanical ball milling [45], cold spraying [46], self-propagating high-temperature synthesis (SHS) [47,48], mechanical mixing [38], and cryomilling [49]. The cryomilling method achieves a relatively high specific surface area compared with the conventional methods. This method also has several advantages compared with room-temperature milling such as the production of small particles without agglomeration and a reduction in the oxidation of powder because the milling process is performed in a nitrogen or argon atmosphere. How to improve the ignition and combustion performance are the major objects for designing the preparation methodology of boron-based composites. Except where specifically pointed out, in this paper, adopted boron means amorphous boron with high reaction activity, which is usually selected as raw material to prepare boron-based composites [50].

### 2.1. Boron–Metal-Based Composites

Compared with the bare boron powder, metals, such as Mg, Al, etc. are easily ignited and can combust fully, making them widely used in solid propellants [51]. Metals or alloys are first adopted to improve the boron combustion performance. The typical structure of amorphous boron is a rough surface and an irregular shape, which can make the particles combine together easily through mutual adhesion, forming stable agglomerated clusters without any external adhesive [2]. The B–Mg-based composite materials were prepared by the typical methodology, which is illustrated in the following. Raw materials with smaller and uniform particle size distribution were preferred to ensure the uniform distribution of magnesium and boron in the agglomerated particles. Agglomerated particles were

prepared by drying the slurry of micron-sized amorphous boron and magnesium powder in ethyl acetate. The B–Mg material was prepared under a vacuum drying environment of 80 °C for 24–48 h. The obtained materials were tested in a simulated real combustion chamber to achieve optical diagnosis and pressure measurement. A gas-driven probe was used to quench and collect the combustion products, and the internal structure evolution of the agglomerated particles in the combustion process was also determined. During the test process, the combustion products were quenched and collected in time by precisely controlling the starting time of the particle feeder and probe. In addition, the collected samples were further analyzed by transmission electron microscopy (TEM). The combustion performance test results of B–Mg binary composites showed that the combustion performances of boron were greatly improved by the introduction of Mg, which easily formed steam and diffused to the surface or caused an explosion of the composite material particles. The steam and the explosion increased the combustion area and provided a fresh surface that eventually enhanced the combustion of boron.

Some refractory metals, such as iron, etc., have also been explored as additives to accelerate boron combustion. The B–Fe based composites can be obtained by high-energy ball milling and surface-coating methods [29]. High-purity iron and amorphous boron powders were selected as raw materials to prepare B–Fe binary composites in a plant ball-milling machine [52]. The B–Fe composites can also be prepared by the thermal decomposition of iron pentacarbonyl [53–56]. The reaction was carried out in a 250 mL glass round bottom flask with a rubber stopper, and the whole device was assembled and operated in a glove box filled with argon. To prepare the B–Fe composite, 4.75 g boron powder was loaded in the vessel with 80 mL dodecane, a 25 mm Teflon®-coated magnetic stirrer (Strider Instruments, Shanghai, China), and 0.5 mL CE-2000 surfactant. The container was rinsed with argon at a rate of 154 mL/min. The exhaust gas was buffered with 400 mL 0.1 M potassium hydroxide solution to capture the decomposition products of the iron pentacarbonyl reaction. The reaction mixture was stirred under argon flow and heated to 110 °C at a rate of 3 °C/min. The temperature was kept constant for one hour to deoxygenate the solution and remove moisture from the powder sample. The temperature was then raised to 190 °C, and 1.5 mL iron pentacarbonyl was slowly introduced into the reaction flask through the diaphragm with a glass syringe. The container was then kept at 190 °C for 2 h for the full decomposition of pentacarbonyl. Finally, the suspension was cooled to ambient temperature and filtered using a Whatman 1003-055 Grade 3 cellulose filter with a diameter of 55 mm (aperture 6 μ m) The powder was collected by vacuum filtration. It was then washed and stored in inert hexane atmosphere. The obtained boron was coated by the decomposed iron, and the structures of the B–Fe composites are shown in Figure 2. It was clearly observed that the boron powder with higher purity had a clear crystal shape and appeared amorphous. Boron with lower purity had amorphous fractal aggregates. There was no significant morphological difference in the SEM test results between boron and iron-coated boron. In addition, a $CO_2$ laser igniter was employed to ignite materials, and a digital camera was used to collect images of particles burning in indoor air. Two photomultiplier tubes (PMTs) equipped with interference filters with central wavelengths of 700 and 800 nm were used to capture the optical emission pulses of the burning particles. Time-resolved spectra of individual particle emission pulses were obtained using a 32-channel Hamamatsu H11460-01 (A10766-007-02) compact spectrometer. The surface-coated iron could replace the boron and react with oxygen. The iron oxides reacted further with boron, which could also inhibit the formation of hydroxyl-boron oxide (HOBO).

In addition to Fe, transition metals, such as Co, Ni, Hf, and Zr, have also been investigated as additives to improve the combustion characteristics of boron [27]. These metals generally have relatively high boiling points, which can ensure their retention during the combustion procedure of boron particles. Boron powder with a purity of 95% was selected as the raw material to prepare boron–transition metal energetic composites by high-energy mechanical milling [27,57]. The percentage of the transition metal in the composites was

5 wt%. A certain amount of boron and metal additives were mixed in a 500 mL steel cylinder. Then, 20 mL Pharmaco (reagent-grade) hexane was added as the process control agent (PCA). The vial was sealed in a glove box filled with argon. Hardened steel balls (3/8″—9.5 mm) were employed as grinding media. After grinding, the samples were recovered at room temperature and dried in an argon atmosphere. The typical images of the starting boron and the prepared composite materials [27]. The obtained results showed that the boron particles were easily aggregated. Each aggregated cluster contained a number of nano-particles. Agglomeration was also observed in the boron–transition metal composites. Compared with single boron powder, the composites displayed a rounder and denser state. Hf and Zr easily reacted with boron and formed metal borides, which can prolong the ignition delay time. On the other hand, the composites of B–Co and B–Ni could not form stable composites, so the combustion of boron was not significantly improved by Co and Ni. The combustion process was recorded by a digital camera to take striped images of luminous particles on the burning sample, and two Hamamatsu R3896-03 photomultiplier tubes (PMTs) (*Hamamatsu* Photonics K.K., Hamamatsu-shi, Japan) were employed to filter light with a wavelength between 700 and 800 nm to record the time-resolved light emission trajectory of a single particle. The combustion of aerosolized powders was also investigated by using a constant volume explosion (CVE) experiment, and the pressure track was recorded by a pressure sensor.

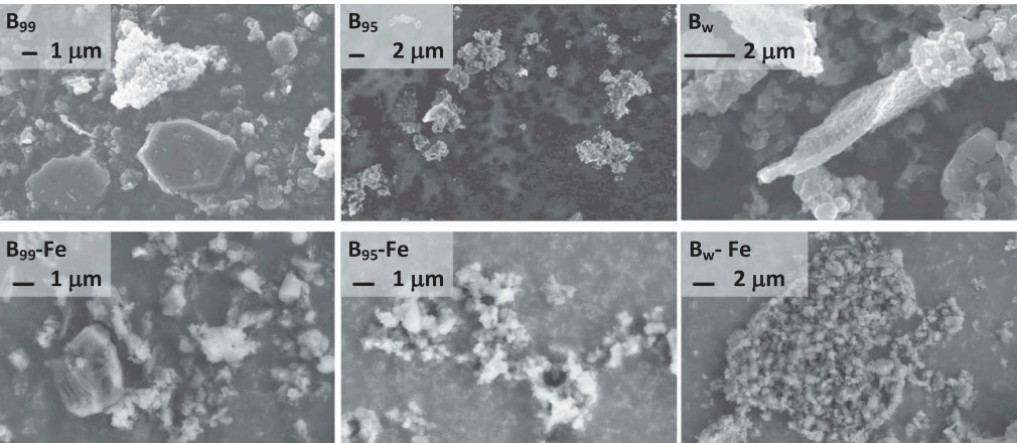

**Figure 2.** SEM images of boron-based materials: Top: Uncoated boron with different purities (secondary electron images); Bottom: Iron-coated boron with different purities and surface modifications (back-scattered electron images). Reprinted with permission from Ref. [52]. Copyright 2021 Taylor & Francis.

### 2.2. Boron–Metal Oxide-Based Composites

Metal oxides can affect the ignition and combustion performance of boron-based thermite. The thermites could be prepared by a facile method. For example, B/nano-NiO thermite composites can be obtained by a simple procedure. First, the nano-Ni(OH)$_2$ was prepared through the reaction of NiCl$_2$·6H$_2$O and NaOH. Then, the prepared nano-Ni(OH)$_2$ was coated on the surface of boron powder by precipitation [32]. After the pyrolysis of B/nano-Ni(OH)$_2$, the B/nano-NiO composite was obtained. XRD tests confirmed that two crystals co-existed in the composite, which indicated that the boron powder was in the amorphous state and the nickel oxide was in a crystalline state. NiO has a high specific surface area and relatively low surface free energy, making it easier for boron to release energy and undergo full combustion. Several synthetic methods of B–metal oxide-based composites had been presented [58]. For example, the B–transition metal oxide composite was synthesized using amorphous boron with a purity of 99.9%, resulting in metal oxides, such as MgO, Al$_2$O$_3$, Bi$_2$O$_3$, CeO$_2$, Fe$_2$O$_3$, CuO, and SnO$_2$, of analytical grade. The B–transition metal oxide composite can also be synthesized by mechanical grinding. Usually,

the transition metal oxides and boron powder are put into a mortar and ground with a pestle for a certain time to ensure full mixing. During the combustion process, with the increase in temperature, the metal oxides are activated and can transfer their oxygen to boron, which would damage the oxide film shell of the boron particles and eventually cause the combustion of the boron particles. The transfer of oxygen from the metal oxide to boron undoubtedly played a positive role in the combustion of boron-based energetic materials.

Other methodologies have been developed to prepare B–metal oxide-based composites that contain CuO, $Bi_2O_3$, and $Fe_2O_3$ [31,59]. Amorphous B powder with a purity of 99.5% and the metal oxide were first ultrasonically treated for 30 min in ethane. Then, the ethane was evaporated, and the precipitate was passed through a 45 micron sieve to obtain the designed composites. The composite containing $MoO_3$ and $Co_3O_4$ was prepared by similar methodology. The SEM test results of the stoichiometric B–metal oxide composites are shown in Figure 3. Figure 3a shows that the raw boron particles had an irregular spherical morphology with submicron size. The SEM of the B–metal oxide composites showed that the amorphous boron was enwrapped by metal oxides. The metal oxide usually had at least one characteristic size less than 200 nm despite different morphologies. Xenon flash ignition experiments were carried out in ambient atmosphere. The ignition delay time and pressure rise of materials under ambient air conditions were quantified in a container with constant volume, where the pressure was recorded by a pressure sensor, and the ignition delay time was calculated from the voltage trace line and the pressure trace line of the pressure sensor. In addition, the combustion heat of the material was measured by an oxygen bomb calorimeter. The metal oxides had higher electron conductivity and oxygen volume density, which caused the diffusion of oxygen to boron, and boron therefore had better a combustion performance.

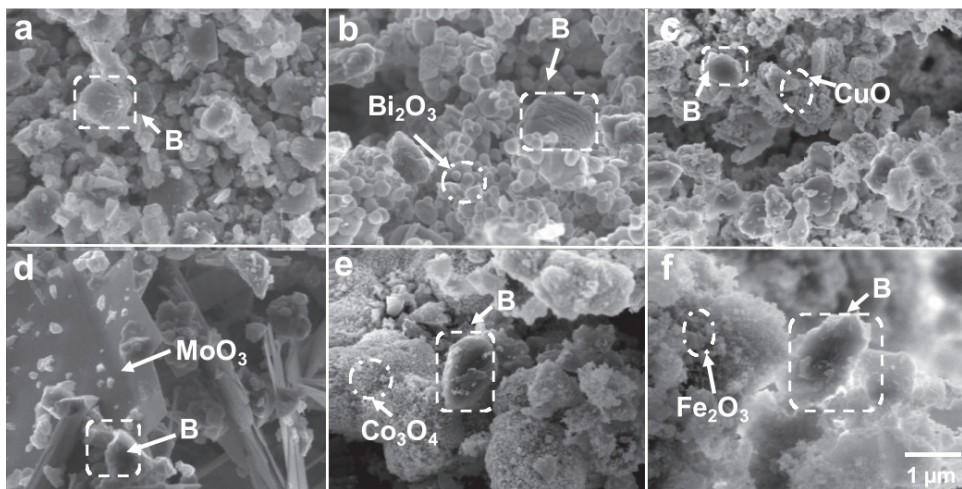

**Figure 3.** SEM of (**a**) boron particles, (**b**) stoichiometric B–Bi2O3, (**c**) B–CuO, (**d**) B–MoO3, (**e**) B–Co3O4, and (**f**) B–Fe2O3. Reprinted with permission from Ref. [31]. Copyright 2019 Elsevier.

### 2.3. Boron-Metal Fluoride Composites

The combustion of boron usually forms $B_2O_3$ in the state of high viscosity at high temperature, which easily adheres to the surface of boron particles, which will eventually inhibit the continuous combustion of boron particles. Therefore, how to quickly remove the $B_2O_3$ formed during combustion becomes the key factor for the continuous combustion of boron particles. Compared with boron oxide, the boron fluoride or fluorine boric oxide that forms during the process of boron combustion were easily from the boron surface. On the basis of thermite theory, it is suggested that the new reaction materials could be designed in which metal fluorides are utilized to substitute the metal oxide to promote the removal of boron combustion products. Metal fluoride-based nano-composites of aluminum, bismuth,

and cobalt, etc., were prepared by mechanical milling [35,37,38]. The starting material was put into a hardened steel grinding bottle in argon atmosphere to prevent oxidation during grinding. Boron pretreated with acetonitrile was used, and the content of boron oxide/boron hydroxide on the surface of boron powder that was washed by this method was low. During the grinding process, a small amount of hexane could be added as the process control agent. The prepared composite was passivated overnight in a glove box rich in argon. In earlier research, the B–LiF composite was first prepared [60]. The amorphous boron powder was coated by LiF through a neutralization deposition method.

The B–metal fluoride composites can be prepared by adsorption reaction milling [38,61]. Amorphous boron with a purity of 95%, $CoF_2$ with a purity of 98%, and $BiF_3$ with a purity of 99% were selected as raw materials to prepare boron–metal fluorinated composite powders through reaction mechanical milling. The obtained materials were characterized by XRD, which proved the appearance of Co (cubic) and Bi (rhombic) in the energetic materials. This could be attributed to the decomposition of metal fluoride during the milling process. It can be deduced that, due to the mechanical milling, the working condition must be adjusted to avoid the decomposition of metal fluoride. The composition of the obtained materials was also examined by scanning electron microscopy (SEM) and energy dispersive spectroscopy (EDS) tests SEM-EDS tests. The materials were embedded in epoxy resin to conduct the SEM tests. The cross-sectional analysis was carried out, and the results are displayed in Figure 4. Figure 4 obviously reveals the surface characteristics and powder particles of cross sections. The particles with a dark gray color were boron, while the bright particles and the coating or inclusions of large particles were heavy metal fluorides. This result clearly proved the successful preparation of boron and two fluoride composites by mechanical milling methodology. TG/DTG tests were carried out on the composite materials under aerobic and anaerobic conditions by thermogravimetry. The ignition temperature of the powder coated on the Ni-Cr wire with a diameter of 0.5 mm was measured. The ignition time of the powder was determined from the high-speed video (taken by the camera), and the light emission was recorded separately by using a photodiode. The sensitivity of the prepared powders to be ignited by ESD was tested by using a 931 model Firing System.

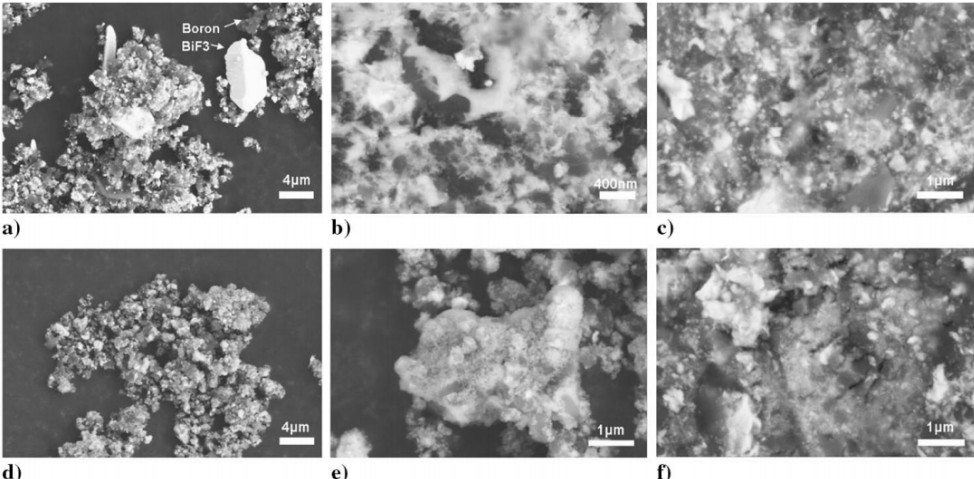

**Figure 4.** SEM images of the 60 min wet-milled composites: (**a**,**b**) B–$BiF_3$ powder as milled, (**c**) B–$BiF_3$ cross-section in epoxy, (**d**,**e**) B–$CoF_2$ powder as milled, and (**f**) B–$CoF_2$ cross-section in epoxy. Reprinted with permission from Ref. [38]. Copyright 2019 AIAA Aerospace Research Central.

The boron–metal fluoride composite can also be prepared by the co-deposition method [35]. For example, the B–$BiF_3$ composites were prepared by the co-precipitation of bismuth nitrate pentahydrate with a purity of 98%, sodium fluoride with a purity of 99%, and amorphous boron powder with a purity of 95% in aqueous media. The B–$BiF_3$ composite was obtained in the solution.

During the combustion procedure of the B–metal fluoride composite, the metal fluoride can produce HF, which can react with the boron oxidant on the surface and form easily evaporated boron fluoride that eventually promotes the ignition and combustion of amorphous boron particles.

### 2.4. Boron–Nano Metal Carbide-Based Composites

Metal carbide-based materials have the advantage of high activity, high combustion heat, a high combustion rate, and high combustion velocity. Under the normal condition, the metal carbide can maintain high stability and decrease the mechanical sensitivity of energetic materials. The metal carbides are usually adopted in the composite of ramjet fuels. The B–nano carbide composites are usually synthesized by mechanical mixing under an argon atmosphere [41]. The high-purity amorphous boron were first mixed with *n*-TiC, *n*-ZrC, and *n*-SiC, separately. Then, the mixtures were ground to ensure the uniformity of the composites. In addition to the metal carbide, other materials such as boron carbide were also adopted to prepare the composites.

The synthesis of the B–$B_4C$ composite has been reported [40,42]. Both boron and $B_4C$ with a purity of 99% were weighed and mixed. Then, the mixture was ground in a planetary mixer under an argon atmosphere for more than 20 min. The obtained composite was the B–$B_4C$ composite. The morphology tests showed that the boron and $B_4C$ were mutually permeated and formed a uniform distribution. The carbides in the composite can produce solid components and carbon dioxide during combustion, which can accelerate the oxidation and break the surface oxide layer of boron and thus improve the combustion efficiency of boron.

### 2.5. Boron–Fluoropolymer-Based Composites

The combustion of boron can be inhibited by the combustion byproducts, which mainly include boron oxides with high viscosity that can prevent continuous combustion. If the boron oxide can be quickly removed from the surface, the combustion of boron can be thoroughly conducted, releasing all combustion heat.

Similar to the combustion characteristics of boron-metal fluoride composites, the combustion of fluoropolymer can also produce HF, which can inhibit the wrapping of boron by the boron oxide and eventually accelerate the combustion of boron-based materials. It was reported that the addition of fluorine accelerated the gasification process of boron oxide, which improved the combustion performance of boron [62]. Compared with metal fluorides, organic fluoropolymers have a better process ability and higher processing safety, which can be employed as formulate additives [23]. The composites of B–PTFE were prepared by a V-type mixer [63]. The boron powder and two kinds of PTFE, namely, 7C and MP-10, were chosen as raw materials. The two materials were uniformly mixed in a V-type mixer and then emulsified thoroughly. The designed materials were obtained after they were fully dried. Similarly, the preparation of the composites of boron with PVDF, Viton, and THV fluoropolymers obtained similar results [23].

The typical procedure for the preparation of the B–fluoropolymer composite is as follows: 1 g B particles was added to a tetrahydrofuran solution containing 40 mg fluoropolymer and ultrasonicated for 20 min to obtain a uniformly dispersed solution. Then, the solution was poured into a Petri dish and kept under a fume hood to evaporate the tetrahydrofuran. The finally obtained powder was collected and fully dried in an oven at 60 °C overnight. Boron powder coated with PVDF, Viton, and THV was prepared by similar methodology. The combustion of boron–fluoropolymer can produce fluoro-alkanes and HF, which are beneficial to remove the boron combustion-formed oxidants instantly. This is the reason for the improvement of boron-based energetic materials that are combined with fluoropolymers.

### 2.6. Boron–Oxidant-Based Composites

The stable combustion of boron needs a continuous supply of oxygen. One of the best oxygen sources for the combustion of boron is sufficient oxidant. Oxidant-enwrapped boron has been investigated as an effective propellant for rockets and other equipment. The oxidants are usually coated on the surface of the boron particles.

Surface coating is a common methodology to prepare boron–oxidant-based composites for metal composites. According to the specific coating material, the oxidant coating can increase the specific surface area, enhance the surface reactivity, and improve the compatibility between particles and binders [43,61]. Two energetic materials, ammonium perchlorate (AP) and nitroguanidine (NQ), are widely utilized as oxidant coating agents [21]. The AP is usually adopted as an inorganic oxidant [64], which has a higher oxygen content and provides sufficient oxygen to enhance the ignition and combustion of boron. The NQ has good surface compatibility, high stability, superior performance, and slight corrosivity [65]. Both of them can provide oxygen to promote the ignition and combustion of boron particles. The typical procedure for the preparation of boron–oxidants is as following. (1) Preparation of the saturated solution of the coating oxidant material, (2) adding a certain proportion of B to the saturated solution, (3) ultrasonicate the mixture for a period, and (4) vaporize and dry the turbid liquid at a certain temperature. The amorphous B with a purity of 99% can be coated by AP or NQ with a mass ratio of 1:10. Boron coated with AP is named BAP, and boron coated with NQ is named BNQ. The AP and NQ decomposition products can break up the surface oxide layer of boron particles and react with the inner boron. The combustion of the composite releases a large amount of heat and greatly shortens the ignition delay time.

The composites of boron with $NH_4ClO_4$, $KNO_3$, $LiClO_4$, and HMX were all prepared by the method [20]. The procedure for the preparation of the B–$KNO_3$ composite was as follows [66]: Firstly, a saturated $KNO_3$ solution was prepared. Then, the amorphous boron with a purity of 99% was added to the solution and ultrasonicated for 15 min. Last, the liquid mixture was evaporated at 60°C. The obtained material uses nitrocellulose as a binder to prepare higher combustion efficiency and higher stability fuels.

On the other hand, the electrostatic spraying method was also developed to prepare boron–oxidant-based spherical composites. The tight contact between the boron and oxidant improved the ignition and combustion performance of the amorphous boron-based materials.

## 3. Combustion Mechanism of Boron-Based Composites

### 3.1. Boron–Metal-Based Composites

The combustion of boron–metal-based composites is greatly influenced by the chemical characteristics of the metal and amorphous boron. The metal and boron have different chemical activities in an oxidant atmosphere. Therefore, the combustion of the composite is greatly influenced by the components. On the other hand, the combustion of the composite is also influenced by the ratio of the compounds. The combustion performance of the B–Mg composites is determined by the content of Mg in the materials. During the combustion process, with the increase in temperature, the oxide layer on the boron particle surface and Mg in the composite begin melting one after another. With the increasing Mg content and rising temperature, the vaporized Mg will diffuse onto the surface of boron through the gap in the material. However, at the same time, the molten Mg can block the diffusion through the internal gap in the B–Mg composite, which results in an agglomeration of generated gaseous Mg inside the composite and eventually leads to the explosion of the particles. The combustion process is finally completed with smaller particles. The low melting point and high vapor pressure of Mg decrease the ignition temperature and shorten the ignition delay time and average combustion time.

Compared with Mg, Fe can also be easily oxidized to form various oxidants, which can play the role of oxygen donor to promote the oxidation of boron particles in B–Fe composites. The reaction between Fe and gaseous oxidants can replace the heterogeneous

reaction between B and gaseous oxidants. On the other hand, the redox reaction between B and $Fe_2O_3$ can also take place in a condensed state, which inhibits the formation of HOBO gas. The combustion of B in B–Fe composites is also influenced by the purity of boron particles. The increased purity of B can enhance the combustion efficiency. On the other hand, regarding the B–Fe composite, with the increase in the Fe content in the composite, the combustion rate can be improved, and the combustion temperature can also be elevated. The existing iron-based oxidant improved the combustion performance of the B–Fe composite.

Compared with iron, heavy metals usually have higher chemical activity, which can easily be ignited and maintain continuous combustion. The B–Hf binary composites have a shorter combustion time than B–Fe composites with the same elemental ratio, which means that the introduction of Hf significantly improve the combustion characteristics of boron. At the same time, it was also observed that the ignition time of B–Hf and B–Zr composites was prolonged, likely because Zr and Hf readily react with B and form their own borides, intermediate products. The formed intermediate usually results in the incomplete combustion of boron. Therefore, the ignition time of B–Hf and B–Zr composites is prolonged. Compared with Hf and Zr, the role of Co and Ni in the improvement of boron-based composite combustion is quite similar to that of Fe, likely because Co and Ni cannot form multiple oxidation states. The boron-based materials can be prepared by mechanical milling accompanied with the modification of particle diameters and surface modification, which can greatly influence the oxidation kinetics of boron powders. Research on the combustion performance of the composites of B–$BiF_3$, B–$CoF_2$, B–Bi, and B–Fe revealed that the combustion kinetics changed considerably with the surface modification, particle size, environment, and metal species.

### 3.2. Boron–Metal Oxide Composites

Metal oxides have been investigated and adopted in practical application as an important component of metal thermites. The metal oxide can react with more active metals and other reducing agents with the release of a large amount of heat. Boron-based thermites form an important branch of thermites. For example, the high combustion performance of NiO-coated B-based thermite has been confirmed. Compared with Mg-PTFE-coated B, the propellant of the B–NiO/Mg-PTFE composite shows enhanced combustion efficiency, a faster combustion rate, and a higher combustion temperature. This is because B–NiO can absorb the heat generated by the meteorological heat flow and condensed phase composition. At the same time, the composite of B–NiO has a higher specific surface and relatively higher surface free energy; therefore, B–NiO can easily react and release a large amount of heat in the combustion procedure. It was also reported that metal oxides can transfer the surrounding oxygen to the B–$B_2O_3$ interface, which can promote the ignition of B and shorten the ignition delay time.

Metal oxides, such as MgO, $Al_2O_3$, $Bi_2O_3$, $CeO_2$, $Fe_2O_3$, CuO, and $SnO_2$, can also dissolve into liquid $B_2O_3$ [67]. With the increase in the surface temperature of B particles, the metal oxide that dissolved into the liquid oxide layer produced mechanical stress at the interface, which broke through the oxide layer and played a positive role in the removal of boron oxide and promoted the combustion of B particles. Due to the different thermal expansion coefficients and the changes in the density during the melting procedure of the composites, the combustion of B particles was promoted. Results showed that the ignition time could be shortened by all metal oxides. $Bi_2O_3$, $Fe_2O_3$, and $SnO_2$ were the most effective metal oxides for the improvement.

Some metal oxides have a relatively high thermal conductivity coefficient, which can enhance the diffusion of oxygen from metal oxides to boron particles. At the same time, the volume density of oxygen in metal oxides is much higher than that of gaseous oxygen. The oxygen can directly contact and diffuse into the core of boron particles, which eventually promotes the combustion of B particles. $Bi_2O_3$ has higher oxygen ion conductivity [68]; therefore, the boron-$Bi_2O_3$ composite can be ignited by the minimum ignition input energy

with the minimum ignition delay time. CuO, MoO$_3$, and Co$_3$O$_4$ can also improve the combustion efficiency of boron particles. CuO has better catalytic performance for the combustion of boron than MoO$_3$ and Co$_3$O$_4$.

### 3.3. Boron–Metal Fluoride Composites

The combustion of boron can form B$_2$O$_3$ and other boron oxides, which are usually in a viscous state at high temperature and easily adhere to the boron matrix. The boron oxide that adheres to boron particles inhibits the diffusion of oxygen to boron and eventually results in the extinction of the combustion flame. During the combustion procedure of boron-based composites, the metal fluoride can react with the organic additives or decompose at high temperature to form HF, which can react with B$_2$O$_3$ and form low viscosity or volatile materials that can easily be removed from the boron surface. The quick removal of high viscosity boron combustion byproducts enhanced the combustion of boron particles, which eventually resulted in the full combustion of boron particles. The ignition temperature of boron-BiF$_3$ and boron-CoF$_2$ composites was lower than the melting point of boron oxide, and the ignition temperature of the composite containing acetonitrile-washed B was higher. The composite prepared with the original boron was easier to ignite than the composite prepared with washed boron. The ignition delay time could also be decreased by adding LiF to the composite. The derived HF removed or eroded B$_2$O$_3$ that grew on the boron surface, which reduced the protective effect of boron oxide. Hydrocarbons and water produced by the decomposition of boron hydroxide provided hydrogen that was necessary for the formation of HF with a low concentration.

### 3.4. Boron–Nano-Carbide Composites

The combustion of boron-containing propellant can generate carbon dioxide, water vapor, and ammonia, which can break the oxide film on the surface of boron particles [20]. It has been proved that metal carbides can be used as effective components of solid propellants [69]. The combustion of metal carbide can generate solid components and carbon dioxide gas. The solid components, i.e., the metal oxide, act as catalysts for accelerating the oxidation and combustion of boron particles, while carbon dioxide destroys the external oxide layer and expands the contact area between boron and oxygen. Therefore, metal carbides can be adopted as additives to promote boron combustion.

At the initial stage of boron particle combustion, the combustion time depends on the temperature and oxidation rate. Low temperature has a negative effect on the evaporation of the surface oxide layer. A high oxidation rate of boron leads to an increase in the output of the oxide layer. The combustion of metal carbide, such as *n*-TiC, *n*-ZrC, and *n*-SiC, releases gaseous substances to remove heat. Among all metal carbides, only *n*-TiC can improve the oxidation rate of boron particles during the course of combustion. Therefore, only n-TiC leads to an increase in the boron oxidation rate at the initial stage of combustion. As boron combustion proceeds, both *n*-TiC and *n*-ZrC have a positive effect on the oxidation rate of boron, which should be attributed to the catalytic role of TiOx and ZrOx in the combustion of boron. The combustion of SiC-produced SiO$_2$ is not active in the oxidation process of boron. n-TiC improves the oxidation efficiency of boron, while n-ZrC prolongs the combustion time. *n*-SiC has no obvious promotion effect on boron combustion. The combustion of boron-based fuels can also be examined by thermogravimetric analysis (TGA), which is usually conducted under an atmosphere of nitrogen, argon, and oxygen. Based on different heating tests, the activation energy can be deduced. There is potential for the application of TGA tests in the evaluation of fuel combustion. More work including theory and experiments should be conducted in future research.

The ignition temperature of *n*-B$_4$C is rather low compared with that of pure boron. The compositional boron is the precedent for the oxidation of B$_4$C, which accelerates the oxidation of boron particles. At the same time, the generated gaseous CO$_2$ of the B$_4$C combustion process promotes the decomposition of the oxide film on the boron particle surface. Therefore, the addition of *n*-B$_4$C reduces the heat absorption of boron combustion

at the initial ignition stage, increases the heat release rate, and accelerates the oxidation of boron particles.

### 3.5. Boron–Fluoropolymer Composites

The combustion of fluoropolymer produces fluorine species, which can react with oxide film on the boron particles that plays an important role in improving the oxidation and combustion characteristics of boron particles. When fluorine atoms are contained in the composites, the removal rate of boron oxide is obviously increased. During the pyrolysis process, THV (polymer of tetrafluoroethylene, hexafluoropropylene, and vinylidene fluoride) produces fluorine-rich alkanes, which greatly enhance the reaction rate of the boron oxide film, which also effectively gasifies the oxide and forms high energy and high temperature in the composites. The combustion reaction activity of the THV composite is greatly enhanced. Compared with Viton, PVDF (polyvinylidene fluoride) also produces a large amount of HF during pyrolysis despite that the reactivity of PVDF with boron oxide is relatively low. The oxidation and combustion characteristics of boron powder can be improved with an increase in the fluorine content of fluoropolymer. THV showed the highest improvement effect on the combustion performances, such as oxidation heat, combustion reactivity, and combustion temperature, during boron combustion. The reactivity of Viton and PVDF follows that of THV, which is consistent with the sequence of the fluorine content in the polymers.

### 3.6. Boron–Oxidant Composites

To fully utilize the energy of fuels, an oxidant, as an oxygen source for fuels, is usually added according to the oxygen index of the designed fuel. As the most widely adopted oxidant in propellants, AP and NQ have their special characteristics, such as low cost, easy production, high stability, and so on. Propellants containing AP and NQ showed better combustion performance than the fuels without them. Especially in the ignition stage, the elevated temperature promotes the decomposition of AP and NQ, which can further accelerate the ignition and combustion of the fuels. With the acceleration of combustion, the coating agent decomposes and produces gaseous materials, which can destroy the oxide film on the surface of the boron particles. On the other hand, the coating oxidant will also release a large amount of heat and further accelerate the gasification rate of boron oxide. In the stable combustion stage, the generated decomposition products of propellants will react with boron, and the intermediate products further promote the combustion of the boron particles. Therefore, compared with pure boron, the average ignition time and average combustion time of the composite propellant containing oxidant decrease. In addition, materials containing NQ also have a higher combustion performance than the materials containing AP, which can be attributed to the high combustion heat of NQ.

Nitrate explosive has also attracted great attention due to its explosive performance, especially as the oxygen source for other fuels. The decomposition of metal nitrate produces metal oxide and nitrogen–oxygen compounds. The nitrogen–oxygen compounds easily ignite the organic fuels and promote combustion. For example, $K_2O$ and $Li_2O$ will be produced in the decomposition process of $KNO_3$ and $LiClO_4$, which can play the role of catalysts in the combustion of boron and decrease the initial reaction temperature. The decomposition of $NH_4ClO_4$ generates gaseous products that easily diffuse onto the surface of boron particles, destroying the surface oxide layer and increasing the contact area between boron and nitrogen–oxygen compounds. The introduction of $LiClO_4$ can greatly decrease the oxidation difficulty of boron. Composites containing $NH_4ClO_4$ have a lower ignition activation energy and better combustion performance. The addition of $KNO_3$ can improve the ignition performance of composite materials [70]. HMX (cyclotetramethylenete–tranitramine) itself has a high weight heating value [71]. Therefore, the composite propellant containing HMX is one of the most promising candidates for high-energy propellants and shows great potential for practical application [72].

## 4. Summary and Outlook

Boron, with a combustion heat of 58 MJ/kg, has attracted great attention as one of the most promising candidates for high-energy propellants. However, the combustion heat of boron is difficult to be fully released due to the characteristics of a high ignition temperature, facile oxidation, incomplete combustion, and combustion products that easily adhere to the particle surface. Many methods have been investigated to promote the full oxidation of boron particles. In this paper, we have reviewed the investigations on the modification of metals, metal oxides, metal fluorides, nano-metal carbides, fluoropolymers, and oxidants with respect to boron to improve the ignition and combustion performances. All of the above-mentioned materials have some effect on the improvement of the ignition and combustion characteristics of boron particles. The improvement of the ignition and combustion of boron particles is mainly aimed at how to effectively eliminate the oxide layer and rapidly remove the combustion product from the surface of the particles. Based on this principle, the metal-based materials with high combustion heat are adopted to improve the ignition and combustion performances. The fluorine-based additives are adopted to enhance the elimination of boron combustion products from the surface of the boron particles. The oxidant provides a large amount of oxygen and can be fully utilized to enhance the combustion efficiency of boron-based materials. The methodology for the preparation of the boron-based composite materials includes mechanical grinding, co-precipitation, ultrasonic mixing, and surface coating. The major purpose of the preparation method is to evenly disperse boron particles with the accessory ingredient and allow the combustion of boron to proceed continuously. Although many methodologies have been investigated for the improvement of boron-based fuel combustion, there is still potential for further enhancement. Opportunities and challenges coexist for the development of boron-based composites with low toxicity and high energy density.

**Author Contributions:** Conceptualization, L.H.; methodology, R.W. and Z.W.; software, W.C. and X.Z.; formal analysis, R.W.; investigation, X.Z.; writing—review and editing, L.H. and T.H.; supervision, Z.W. and T.H. All authors have read and agreed to the published version of the manuscript.

**Funding:** We are grateful for the financial support from the Jinan Science and Technology Development Plan, grant number 202019165, Shandong Provincial Natural Science Foundation, grant number ZR2021MH112, ZR2021MB020, ZR2020MB089 and ZR2020MH044, National Natural Science Foundation of China, grant number 22075290.

**Conflicts of Interest:** The authors declare no conflict of interest.

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
