# Peer review of "Preparation and Combustion Mechanism of Boron-Based High-Energy Fuels"

_catalysts, doi:10.3390/catal13020378_

Round 1

Reviewer 1 Report

This paper reports the preparation and combustion mechanism of boron based high energy fuels, this work could be useful towards alternative propulsion technologies, whereas technical English has to be improved.

present some beneficial advices on how to promote the preparation of boron based, Justification and innovation details still to be improved, need to be emphasized

In figure 5, it is mentioned that SEM image is given, but when we look at referances, it is understood that EDS mapping is done.  It would be more accurate to visualize it as SEM-EDS mapping. 

Boron ignition mechanism could be described

“The composition of the obtained materials were also examined by SEM tests”  SEM imaging does not provide information about composition. 

Not only the characterization of boron-containing materials, but also thermogravimetric analysis should be considered in more detail.

Author Response

Response:

Thanks for the reviewer’s work on our manuscript. It could be seen that the reviewer is an expert on materials’ test technology. We have revised the corresponding content and the revisions are listed as following.

Page 7 Line 1:

The composition of the obtained materials were also examined by SEM-EDS tests.

Page 11 Line 25:

The combustion of boron based fuels can also be examined by thermogravimetric analysis (TGA), which are usually conducted under the atmosphere of nitrogen, argon and oxygen. Based on different heating tests, the activation energy can be deduced. Until now, there still large space for the application of TGA tests in the evaluation of fuels combustion. More work including theory and experiments should be done in the following researches.

Reviewer 2 Report

Interesting look t boron combustion and trying to understand the combustion mechanisms for boron combustion aids. The paper is a bit speculative as there was a lack of supporting data for their claim. Also, they replicate a  lot of work by Ed Dreizin's group with the milling and addition of the metals, metal fluorides and metal oxides, and would like to know how their data compares.

Expand more on the sample preparation and testing of modeling.  

Author Response

Response:

Thanks for the reviewer’s careful examination on the manuscript. In this paper, the combustion properties of different materials in the references are compared and analyzed, such as ignition temperature, ignition delay time and average combustion time of composite materials. The composite fuel with low ignition temperature, short ignition delay time and average combustion time has better combustion performance. We have revised the corresponding content in the paper.

Page 9 Line 50:

The boron based materials can be prepared by mechanical milling accompanied with the modification of particle diameters and surface modification, which can greatly influence the oxidation kinetics of boron powders. The composites of boron-transition metal and boron compounds, etc. Researches on the combustion performance of the composites of B-BiF3, B-CoF2, B-Bi, B-Fe revealed that the combustion kinetics was greatly changed with the surface modification, particle sizes, environments and metal species.

Reviewer 3 Report

In this study, the authors review the methods of the preparation of boron-based fuels and the combustion mechanism of boron based composite. The manuscript is well written, concise and the authors have nicely presented the impressive amount of data from literature. The manuscript is publishable after the below mentioned questions/comments are addressed. I recommend "accept after minor revision".

Comments

1) What is the temperature for evaporating the B2O3 from the boron particle surface?

2) The cryomilling method is one of the useful methods for preparation of composite powders. This method achieves a relatively high specific surface area compared with the conventional methods. This method also has several advantages compared with room temperature milling such as the production of small particles without agglomeration and reducing the oxidation of powder because the milling process is performed in a nitrogen or argon atmosphere. Please added this method for preparation of boron composite powders to the manuscript text (Materials 2022, 15(13), 4618; https://doi.org/10.3390/ma15134618).   

Author Response

Comments

1) What is the temperature for evaporating the B2O3 from the boron particle surface?

Response:

Thanks for the reviewer’s careful examination. The evaporation temperature of B2O3 is 1860 ℃ and it is also supplied in the manuscript Page 2 Line 4.

Page 2 Line 4:

As shown in Figure 1, the model fully took into account the evaporation process of B2O3, which can be happened at 1860 ℃, and B2O2.

Comments:

2) The cryomilling method is one of the useful methods for preparation of composite powders. This method achieves a relatively high specific surface area compared with the conventional methods. This method also has several advantages compared with room temperature milling such as the production of small particles without agglomeration and reducing the oxidation of powder because the milling process is performed in a nitrogen or argon atmosphere. Please added this method for preparation of boron composite powders to the manuscript text (Materials 2022, 15(13), 4618; ).

Response:

Thanks for the reviewer’s constructive comments. It could be seen that the reviewer is expertized in materials preparation methods. We have added the cryomilling method to our manuscript. The mentioned reference is also cited as reference 49.

Page 3 line 41:

and cryomilling method [49]. The cryomilling method achieves a relatively high specific surface area compared with the conventional methods. This method also has several advantages compared with room temperature milling such as the production of small particles without agglomeration and reducing the oxidation of powder because the milling process is performed in a nitrogen or argon atmosphere.